# The GBM Tumor Microenvironment as a Modulator of Therapy Response: ADAM8 Causes Tumor Infiltration of Tams through HB-EGF/EGFR-Mediated CCL2 Expression and Overcomes TMZ Chemosensitization in Glioblastoma

**DOI:** 10.3390/cancers14194910

**Published:** 2022-10-07

**Authors:** Xiaojin Liu, Yimin Huang, Yiwei Qi, Shiqiang Wu, Feng Hu, Junwen Wang, Kai Shu, Huaqiu Zhang, Jörg W. Bartsch, Christopher Nimsky, Fangyong Dong, Ting Lei

**Affiliations:** 1Sino-German Neuro-Oncology Molecular Laboratory, Department of Neurosurgery, Tongji Hospital, Tongji Medical College, Huazhong University of Science and Technology, Wuhan 430030, China; 2Department of Neurosurgery, Philipps University Marburg, Baldingerstrasse, 35033 Marburg, Germany

**Keywords:** glioblastoma, temozolomide, tumor-associated macrophages/microglia, CCL2, EGFR signaling, ADAM8, chemoresistance

## Abstract

**Simple Summary:**

Resistance to standard therapies impose a huge challenge on the treatment for glioblastoma multiforme (GBM), which is often considered as a cell intrinsic property of either GBM or, more significantly, of GBM stem-like cells. Tumor-associated macrophages and microglia (TAMs) take up the majority of the immune population in the tumor microenvironment of GBM and potentially participating in modulating therapy responses. However, little is known about the mechanisms underlying the effect of TAMs on temozolomide (TMZ) induced chemoresistance. Members of the metzincin superfamily such as Matrix Metalloproteases (MMPs) and A Disintegrin and Metalloprotease (ADAM) proteases are important participants in the process of intercellular communications in the tumor microenvironment. Herein, we revealed a novel concept of an intra-tumoral ADAM8 mediated malignant positive feedback loop constituted by the intimate interaction of tumor associated macrophages (TAMs) and GBM cells under TMZ treatment. These findings provide a convincing example and further support the notion that the tumor microenvironment, in addition to GBM cells and GBM stem-like cells, should be considered as an essential modulator of therapy in GBM. In conclusion, our study provides a rational basis for TAM sparing ADAM8-targeting in GBM to optimize standard chemotherapy.

**Abstract:**

Standard chemotherapy of Glioblastoma multiforme (GBM) using temozolomide (TMZ) frequently fails due to acquired chemoresistance. Tumor-associated macrophages and microglia (TAMs) as major immune cell population in the tumor microenvironment are potential modulators of TMZ response. However; little is known about how TAMs participate in TMZ induced chemoresistance. Members of the metzincin superfamily such as Matrix Metalloproteases (MMPs) and A Disintegrin and Metalloprotease (ADAM) proteases are important mediators of cellular communication in the tumor microenvironment. A qPCR screening was performed to identify potential targets within the ADAM and MMP family members in GBM cells. In co-culture with macrophages ADAM8 was the only signature gene up-regulated in GBM cells induced by macrophages under TMZ treatment. The relationship between ADAM8 expression and TAM infiltration in GBM was determined in a patient cohort by qPCR; IF; and IHC staining and TCGA data analysis. Moreover; RNA-seq was carried out to identify the potential targets regulated by ADAM8. CCL2 expression levels were determined by qPCR; Western blot; IF; and ELISA. Utilizing qPCR; IF; and IHC staining; we observed a positive relationship between ADAM8 expression and TAMs infiltration level in GBM patient tissues. Furthermore; ADAM8 induced TAMs recruitment in vitro and in vivo. Mechanistically; we revealed that ADAM8 activated HB-EGF/EGFR signaling and subsequently up-regulated production of CCL2 in GBM cells in the presence of TMZ treatment; promoting TAMs recruitment; which further induced ADAM8 expression in GBM cells to mediate TMZ chemoresistance. Thus; we revealed an ADAM8 dependent positive feedback loop between TAMs and GBM cells under TMZ treatment which involves CCL2 and EGFR signaling to cause TMZ resistance in GBM.

## 1. Introduction

Glioblastoma multiforme (GBM) is an extremely malignant central nervous system tumor with an annual incidence rate of 3–5/100,000 and a dismal prognosis of 14.6 months survival, accounting for about 50% of all gliomas [1,2]. Multimodal treatment incorporating surgical resection, radiotherapy, and chemotherapy is the standard regimen for GBM patients [3]. Temozolomide (TMZ), the first-line alkylating agent for GBM chemotherapy [4], efficiently penetrates the Blood-Brain Barrier (BBB) and causes cytotoxicity by inducing DNA double-strand damage. In previous studies, many genes related to DNA damage repair played an important role in drug resistance, such as alkylpurine-DNA-N-glycosidase, which may be the cause of TMZ resistance in glioblastoma [5,6,7]. However, TMZ efficacy has not been improved for GBM patients over the past 10 years, indicating that the underlying mechanism of TMZ resistance needs to be further excavated.

Both intrinsic characteristics of cancer cells and extrinsic interactions within the sophisticated tumor microenvironment (TME) contribute to treatment resistance and tumor aggression [8]. Increasing evidence manifested that chronic inflammation in TME was closely related to cancer initiation, promotion, and progression. Macrophages are the dominant orchestrators of tumor-promoting inflammatory signals and a high density of tumor-associated macrophages (TAMs) infiltration is associated with high-grade tumors and a dismal prognosis [9]. In GBM, TAMs (resident microglia and bone marrow derived-macrophages) constitute the majority of inflammatory cells, accounting for up to 40% of the bulk tumor. In general, TAMs present as an immunosuppressive tumor-supportive phenotype and play significant roles in tumor proliferation, migration, and invasion [10]. Mitchem et al. found that TAMs promoted the chemotherapy resistance of pancreatic cancer and the regional immunosuppressive response [11]. Wang et al. demonstrated that TAMs locally aggregated in gliomas can promote tumor growth [12]. Hu et al. confirmed that TAMs can promote the invasion and tumor expansion of malignant gliomas [13]. Zhou et al. also found that TAMs recruited by glioblastoma stem cells promoted malignant growth [14]. However, little is known about the role of TAMs in promoting tumor chemoresistance to alkylating agents in GBM.

A disintegrin and metalloproteinase 8 (ADAM8) is a transmembrane protein consisting of 856 amino acids, involved in many physiological functions such as cell adhesion, cell fusion, signal transduction, and proteolysis [15]. ADAM8 expression in the central nervous system (CNS) is very low under physiological conditions. However, in the case of CNS inflammation, i.e., caused by increased expression of tumor necrosis factor alpha (TNF-a), the expression levels of ADAM8 in astrocytes and microglia are significantly increased [16], and overexpressed ADAM8 can promote the local matrix remodeling through proteolysis or cleavage of other substrates [17,18]. In addition, the markedly increased ADAM8 in a variety of malignant tumors has also attracted attention to its role in the malignant behavior of tumors [15]. Wildeboer et al. [19] showed increased invasion of ADAM8 expressing GBM cells. He et al. [20] showed that ADAM8 expression was significantly related to tumor progression and patient prognosis. Li et al. demonstrated that ADAM8 affected angiogenesis in GBM [21]. Considering ADAM8 as a protein that plays an important role in both CNS inflammation and tumor pathology, we reasonably believe that ADAM8 may mediate the malignant biology of tumor cells through enhancing or maintaining local inflammatory responses in GBM. Studies have shown that some members of the ADAM family (ADAM10, ADAM12, and ADAM17) participated in the ectodomain shedding of the EGFR ligand and mediated the EGFR signaling pathways in certain circumstances [22]. However, it is not clear whether ADAM8 is involved in the EGFR signaling pathways in GBM under TMZ treatment. An et al. validated that EGFR cooperated with EGFRvIII to induce CCL2-mediated macrophages recruitment [23]. In our previous study, we validated that TMZ induced ADAM8 overexpression in GBM cells [24]. Therefore, we hypothesize that ADAM8 may induce TAMs recruitment through EGFR signaling-mediated CCL2 expression in GBM under TMZ treatment to induce chemoresistance.

In this study, we investigated the role of ADAM8 in recruiting TAMs to mediate chemoresistance and put forward a potential ADAM8 positive feedback loop involved in the chemo-resistance between TAMs and GBM cells, providing a theoretical basis for ADAM8-targeting treatment of GBM patients in the future.

## 2. Materials and Methods

### 2.1. Patient Specimens

Tumor tissues of patients who underwent surgical resections of GBM were collected after patients provided informed consent at Tongji Hospital of Huazhong University of Science and Technology. Fresh tissues were immediately snap frozen in liquid nitrogen and preserved at −80 °C until RNA separation or embedded in paraffin for immunohistochemistry and immunofluorescent staining. The Human Ethics Committee of Tongji Hospital of Huazhong University of Science and Technology approved this study, and all the studies were in accordance with the ethical standards of the 2008 Helsinki Declaration.

### 2.2. Cell Lines, Cell Culture, and Construction of Stable Cell Lines

Established human glioblastoma cell lines U87MG and U251MG and THP-1 monocyte were purchased from the American Type Culture Collection and provided by Sino-German Neuro-Oncology Molecular Laboratory. GBM primary cells (G1 and G2) were prepared from human GBM specimens collected directly after surgery as described [24]. To generate glioblastoma cells (U87MG and G1) with an ADAM8 knockdown, the small hairpin RNA (shRNA) of human ADAM8 (target sequences: CGTGGACAAGCTATATCAGAA and GCATGACAACGTACAGCTCAT) were synthesized and cloned into the PLKO.1-puro vector (Invitrogen, Chongqing, China). ADAM8 knockdown and control plasmids were co-transfected into HEK293T cells with pMD2.G and psPAX2 plasmids using lipofectamine 3000 (Invitrogen, Waltham, MA, USA) following the manufacturer’s instructions, respectively. After 72 h, the medium supernatant was collected and the virus liquid was obtained after 0.45 μm filtration. Glioblastoma cells were seeded into 6-well plates and cultured with virus liquid. After lentivirus transfection for 3 days, Puromycin was added to screen for transformants. After 14 days selection, stable ADAM8 knockdown GBM cell lines (U87MG and G1) were obtained. ADAM8 expression in single cell clones was analyzed by qRT-PCR and western blotting. GBM cell lines were cultured in DMEM high glucose (4.5 g/L) supplemented with 1% L-glutamine (200 mM), 1% penicillin/streptomycin, and 10% fetal bovine serum (heat-inactivated) (all purchased from Gibco company, NY, USA). THP-1 cells were maintained in RPMI 1640 medium containing 10% Fetal Bovine Serum (FBS) at 37 °C in a humidified atmosphere with 5% CO_2_. THP-1 cells were primed with PMA (Sigma (Saint Louis, MI, USA), 100 ng/mL) for 48 h to generate unpolarized macrophages (M0) as macrophage model cell line.

### 2.3. Antibodies and Agents

Primary antibodies against ADAM8, EGFR, CCL2, and Akt (panAkt and pAkt S473) were obtained from Proteintech; p-EGFR, ERK1/2 (panERK and pERK1/2 Tyr 202/204) from Cell Signaling Technology (Danvers, MA, USA); HB-EGF from ABclonal technology (Woburn, MA, USA). Iba-1, GFAP, and CD206 were purchased from Abcam (Cambridge, UK). Secondary antibodies were listed in the Appendix A together with primary antibodies. Temozolomide was purchased from Selleckchem (Houston, TX, USA). As an EGFR inhibitor, we used Erlotinib (Selleckchem).

### 2.4. Quantitative Real-Time PCR (qRT-PCR)

Total RNA was isolated from GBM tissues and cell lines using Trizol RNA isolation reagent (Invitrogen) and reversely transcribed to cDNA with a cDNA Synthesis kit (Yeasen Biotech Co., Shanghai, China). qRT-PCR was used to detect the gene expressions with Hieff^®^ qPCR SYBR Green Master Mix (Low Rox Plus) (Yeasen Biotech Co. China). The sequences of primers are shown in Appendix A.

### 2.5. ELISA

Human CCL2 levels in cell culture supernatants were determined by ELISA kits for human CCL2 (Elabscience, Houston, TX, USA) according to the manufacturer’s instructions.

### 2.6. Cell Proliferation Assay

Co-cultures of GBM cells with macrophages were performed in the presence of TMZ. In brief, THP-1 derived macrophages were seeded into a 6-well transwell chamber with an 0.4 μm pore size polycarbonate membrane (Corning, NY, USA) at a density of 2 × 10^5^ cells, and GBM cells were seeded in the lower chamber at a density of 2 × 10^5^ cells in the presence of TMZ. After 3 days, GBM cells were fixed with 4% paraformaldehyde and Crystal Violet Staining Solution (Servicebio, Wuhan, China) was used to visualize the cells. We randomly selected five visual fields of each group under the microscope and the number of GBM cells were measured by ImageJ software.

### 2.7. Cell Migration and Invasion Assays

Co-Cultures of GBM cells with THP-1 derived macrophages were performed in transwell assays. For the macrophage migration assay, THP-1-derived macrophages were seeded into a 24-well transwell chamber with an 8 μm pore size polycarbonate membrane (Corning, NY, USA) at a density of 5 × 10^4^ cells in 200 μL serum-free medium, TMZ (500 μmol/L, 5 days)—or DMSO-treated U87MG_scramble, U87MG_shA8, G1_scramble cells, and G1_shA8 were seeded into the lower chamber at a density of 5 × 10^4^ cells in 400 μL 10% FBS medium. For the GBM cell migration assay, GBM cells were seeded into the 24-well transwell chamber at a density of 2 × 10^4^ cells, and THP-1-derived macrophages were added to the lower chamber at a density of 5 × 10^4^ cells. After 24 h for GBM cells and 48 h for THP-1 derived macrophages, 4% paraformaldehyde was used to fix the migrated cells and Crystal Violet Staining Solution (Servicebio) was used to visualize the cells. For GBM cells invasion assay, 3 × 10^4^ cells were maintained in the matrigel (BD Bioscience, San Jose, CA, USA) coated chamber for 24 h. Three independent experiments were carried out and analyzed through ImageJ software.

### 2.8. Immunofluorescence Staining

Immunofluorescent staining was performed in GBM xenografts and human GBM tissues. Primary antibodies included IBA-1(1:500, Abcam), GFAP (1:200, Abcam), p-EGFR (1:200, Cell Signaling Technology), p-ERK (1:200, Cell Signaling Technology), HB-EGF (1:100, Abclonal Technology), CCL2 (1:200, Proteintech, Rosemont, IL, USA), Briefly, tumor sections were deparaffinized, rehydrated, and antigen-retrieved by standard procedures. Then, samples were blocked with a PBS solution containing 1% BSA plus 0.3% Triton X-100 for 2 h at room temperature and then incubated with indicated primary antibody overnight at 4 °C followed by the fluorescence-conjugated second antibody (1:200) at room temperature for 2 h. After being counterstained with DAPI for 5 min, sections were mounted on glass and subjected to microscopy. The extent of positive cells was measured by ImageJ, which was defined as the ratio of positive cells relative to the total cells in five randomly selected viewing fields. The images were acquired with a fluorescent microscope (Olympus (Tokyo, Japan), CKX53).

### 2.9. Immunohistochemistry Staining

Immunohistochemistry staining was performed as described in our previous publication [25]. Primary antibodies included ADAM8 (1:100, Proteintech), IBA-1 (1:500, Abcam), CD206 (1:1000, Abcam). The percentage of positive cells or the percentage of positive areas was calculated in five randomly selected fields using ImageJ software (ImageJ 1.8.0, NIH, Bethesda, MD, USA) respectively.

### 2.10. Xenograft Studies in Nude Mice

Human U87MG_scramble and U87MG_shA8 cells (5 × 10^6^ in 100 μL PBS) were inoculated subcutaneously into the right armpit of BALB/c nude mice (6-week-old, male). After 7 days, the mice bearing tumors were randomized into U87MG_scramble + drug vehicle (PBS and dimethyl sulfoxide [DMSO]) group, U87MG_scramble + TMZ group, U87MG_shA8 + drug vehicle group, and U87MG_shA8 + TMZ group. Mice in TMZ groups received 5 mg/kg TMZ on 5 days on/2 days off regimen (two cycles in total, intranodal injection.), and mice in drug vehicle groups received equivalent volumes of drug vehicle. About 28 days after the first treatment, all mice were euthanized, and the tumor masses were carefully removed, measured, and processed for IHC and IF staining. For in-site inoculation, five thousand human U87MG_scramble and U87MG_shA8 cells were transplanted into the right frontal lobe of mice. After 7 days, the mice bearing tumor received 5 mg/kg TMZ on 5 days on/2 days off regimen (two cycles in total, intraperitoneal injection.). Overall survival was analyzed between the U87MG_scramble and U87MG_shA8 groups.

### 2.11. Western Blot

Total protein was extracted with RIPA buffer, and 20–50 μg samples were loaded after measuring their concentration using a BCA kit and separated by 6%, 8% or 12% sodium dodecyl sulfate-polyacrylamide gel electrophoresis. The separated proteins were transferred onto polyvinylidene fluoride membrane blocked with 5% fat-free milk for 2 h at room temperature and incubated in primary antibodies against ADAM8 (1:1000, Proteintech), HB-EGF (1:1000, Abclonal Technology), EGFR (1:1000, Proteintech), p-EGFR (1:1000, CST, Beverly, MA, USA), ERK (1:1000, CST), p-ERK (1:1000, CST), AKT (1:1000, Proteintech), p-AKT (1:1000, Proteintech), CCL2 (1:1000, Proteintech), and GAPDH (1:2000, Enogene, New York, NY, USA) at 4 °C overnight. Membranes were washed and incubated in horseradish peroxidase (HRP)-conjugated secondary antibodies for 1 h at room temperature. An enhanced chemiluminescence system (NCM Biotech, Suzhou, China) was used to detect the protein bands.

### 2.12. RNA—Seq Data Analysis

U87MG_shA8 and U87MG_scramble cells were harvested in TRizol for RNA extraction and sequencing by BGI (Beijing Genomic Institute in Shenzhen). Briefly, SOAPnuke (v1.5.2, BGI, Shenzhen, China) was used to filter the sequencing data, then clean reads were stored in FASTQ format, mapped to the reference genome utilizing HISAT2 (v2.0.4, Hopkins, Baltimore, MD, USA). Afterwards, fusion genes and differential splicing genes (DSGs) were analyzed through Ericscript (v0.5.5), and rMATS (V3.2.5, Sourceforge, San Diego, CA, USA), respectively. The clean reads were aligned by Bowtie2 (v2.2.5, Hopkins, Baltimore, MD, USA) to a known and novel database built by BGI, which includes coding transcript, then RSEM (v1.2.12) was applied to calculate the levels of gene expression. Differential expression was analyzed using the DESeq2 (v1.4.5, UNC, Chapel Hill, NC, USA) with a *Q* value ≤ 0.05. To deduce the phenotype change, a GO and KEGG enrichment analysis was performed by Phyper in the basis of Hypergeometric test. Bonferroni was used to correct the significant levels of terms and pathways by *Q* value with a rigorous threshold (*Q* value ≤ 0.05)

### 2.13. Statistical Analysis

An unpaired two-tailed Student’s *t*-test and Pearson’s Χ2-test were used to analyze the variances in each experimental group. The Kaplan–Meier method and log-rank test were used to estimate survival probabilities. Based on the obtained results, the data were considered not significant (*p* > 0.05), significant (* *p* < 0.05, ** *p* < 0.01, *** *p* < 0.001 and **** *p*< 0.0001). Calculations were performed using the GraphPrism statistical analysis software (v6.0, GraphPad Software Inc., CA, USA).

## 3. Results

### 3.1. ADAM8 Expression Induced by TMZ and Macrophage Co-Culture

Initially, we performed a qPCR screen to analyze expression levels of *MMP* and *ADAM* genes in GBM cell lines (U87MG and U251MG) and primary cells (G1 and G2) under TMZ treatment (500 ng/mL) and co-culture with THP-1 derived macrophages for 3 days (Appendix A). Among all MMPs and ADAMs detected, we found that ADAM8 was significantly upregulated in GBM cells by TMZ, particularly under conditions of co-culture (Figure 1A–D). Western blot analyses were carried out to further validate TMZ and macrophage-induced ADAM8 overexpression in GBM cells (Figure 1E–H; The uncropped Western blots have been shown in Appendix A).

### 3.2. Positive Correlation of ADAM8 Expression and Macrophages Infiltration in GBM Tissue

To investigate the relationship between ADAM8 expression and macrophage infiltration level, bioinformatic analyses of correlation using the public dataset GEPIA (Gene Expression Profiling Interactive Analysis) were performed. We selected GBM samples from TCGA projects on GEPIA and found a positive correlation between ADAM8 gene expression and expression of TAM signatures including Iba-1 (AIF1, R = 0.38, *p* = 4.5 × 10^−7^), CD11b (ITGAM, R = 0.63, *p* = 0), CD163 (R = 0.56, *p* = 4.9 × 10^−15^), and CD206 (MRC1, R = 0.64, *p* = 0) (Figure 2A), indicating that ADAM8 expression is associated with TAMs and may play a role in attracting TAMs into GBM. Using our GBM patient cohort (*n* = 18), the previous positive relationship between mRNA levels of ADAM8 and Iba-1 (Figure 2A) was validated by immunostaining and qPCR (*N* = 18, R^2^ = 0.5316, *p* = 0.0006) (Figure 2B,C). Moreover, immunohistochemistry staining of GBM tissues showed that high-ADAM8 expression groups tend to have a higher density of infiltrated TAMs than the low-ADAM8 expression group (Figure 2D,E).

### 3.3. ADAM8 Induces Macrophage Recruitment In Vitro and In Vivo

To investigate the role of ADAM8 in TAMs recruitment in vitro, we constructed ADAM8 knockdown GBM cells (U87MG_shA8 and G1_shA8) and scramble controls (U87MG_scramble and G1_scramble) and co-cultured these cells with THP-1 derived macrophages using a standard protocol to obtain M0 macrophages. THP-1 derived macrophages were seeded into 24-well transwell chambers at a density of 5 × 10^4^ cells in 200 μL serum-free medium. TMZ- (500 μmol/L, 5 days) or DMSO-treated U87MG_shA8, G1 and shA8 and U87MG_scramble and G1_scramble cells were seeded into the lower chamber at a density of 1 × 10^4^ cells in 400 μL 10% FBS medium for 48 h. For the group of TMZ-treated GBM cells, significant increases in the number of migrated macrophages were observed. Compared to scramble controls, both ADAM8 knockdown GBM cells markedly decreased the numbers of migrated macrophages (Figure 3A–D). To evaluate ADAM8-dependent macrophage recruitment in vivo, U87MG_shA8 and U87MG_scramble cells were inoculated subcutaneously into the right flank of BALB/c nude mice (6-week-old, male). After 7 days, the mice bearing tumors were randomized into U87MG_scramble + drug vehicle (PBS and dimethyl sulfoxide [DMSO]) group, U87MG_scramble + TMZ group, U87MG_shA8 + drug vehicle group, and U87MG_shA8 + TMZ group. Mice in TMZ groups received daily 5 mg/kg TMZ for a 5 days on and 2 days off regimen (two cycles in total, intra-tumoral injection), and mice in drug vehicle groups received the equivalent drug vehicle. Twenty-eight days after the first treatment, the tumors were collected and processed. IHC staining of the resulting tumors showed a significantly higher expression of ADAM8, Iba-1, and CD206 in the group of U87MG_scramble + TMZ, compared to U87MG_scramble + drug vehicle. In accordance, ADAM8 knock-down showed a markedly decreased expression of ADAM8, Iba-1 and CD206 (Figure 3E,F), indicating that ADAM8 promotes TAMs recruitment in the presence of TMZ. Moreover, we showed that macrophage promoted GBM cells proliferation, migration and invasion in vitro in the presence of TMZ, indicating that macrophage induced GBM chemoresistance in vitro (Appendix A).

### 3.4. ADAM8 Regulates the HB-EGF/EGFR Signal Pathway

To further analyze the underlying mechanisms of ADAM8-mediated TAMs recruitment, RNA sequencing was performed on U87MG_shA8, and U87MG_scramble cells. We identified 2533 up-regulated and 2855 down-regulated genes in U87MG cells (Appendix A). Then we selected for the top 200 down-regulated genes to undergo GO enrichment to search for the involvement of potential cell signaling pathways. The top twenty GO enrichments are listed in Figure 4A. We observed that among seven genes (MAPK1, MAP2K1, E2F1, CCND1, HBEGF, MYC and VEGFA) in the most significant GO terms, HB-EGF accounts for the most significant signature after ADAM8 knock down (Figure 4B). To further corroborate this finding, qPCR analyses were carried out in GBM cells (U87MG and G1) and revealed that ADAM8 knockdown markedly reduced the expression of HB-EGF (Figure 4C). It has been reported that HB-EGF binds to EGFR thereby activating downstream EGFR signaling cascades including (PI3K/AKT, MAPK/ERK, and JAK/STAT etc.) [26]. Consequently, we determined the protein expression of HB-EGF, p-EGFR, p-AKT, and p-ERK in GBM cells by western blot and observed that ADAM8 knockdown reduced the expression of HB-EGF, p-EGFR, p-AKT, and p-ERK (Figure 4D,E). Moreover, TMZ treatment augmented ADAM8, HB-EGF, p-EGFR, p-AKT, and p-ERK expression (Figure 4D,E). Immunofluorescence staining of xenograft tissue sections showed a significantly higher expression of HB-EGF (Figure 4F), p-EGFR (Figure 4G), and p-ERK (Appendix A) in the U87MG_scramble +TMZ group compared to the U87MG scramble + drug vehicle group. ADAM8 knockdown significantly decreased the expression of HB-EGF (Figure 4F), p-EGFR (Figure 4G), and p-ERK (Appendix A), as manifested in the group of U87MG_shA8 + drug vehicle and U87MG_shA8 + TMZ. Concomitantly, the immunofluorescence of human GBM tissues showed a stronger staining density of HB-EGF in the specimens from high-ADAM8 expression patients compared to the low-ADAM8 expression patients (Appendix A). Taken together, our above data suggest that ADAM8 regulates the HB-EGF/EGFR signaling pathway by affecting expression levels of HB-EGF.

### 3.5. ADAM8 Induce HB-EGF/EGFR Mediated CCL2 Expression

Since the EGFR signaling pathway has been reported to induce CCL2 expression to recruit macrophages in glioblastoma [23], we propose that ADAM8 could induce TAM recruitment by regulating CCL2 expression via HB-EGF/EGFR. Accordingly, qPCR and Western blot showed ADAM8 knockdown reducing intracellular CCL2 expression in GBM cells both at the transcriptional and translational level (Figure 5A). Given the fact that CCL2 is a secreted protein, we applied ELISA to detect the secreted CCL2 in the supernatant of ADAM8 knockdown and TMZ-treated GBM cells. ELISA assays showed that ADAM8 knockdown significantly decreased CCL2 secretion, and TMZ induced the expression of CCL2 in GBM cells. (Figure 5B). Immunofluorescence staining of xenograft tissue sections showed a markedly higher CCL2 expression in the U87MG_scramble +TMZ group compared to the U87MG_scramble + drug vehicle group. *ADAM8* knockdown markedly reduced CCL2 expression (Figure 5C). Moreover, the immunofluorescence of human GBM tissues showed a higher staining density of CCL2 in the high-ADAM8 expression patient group compared to the low-ADAM8 expression patient group (Appendix A). To further validate ADAM8 induced CCL2 expression through the HB-EGF/EGFR signaling pathway, we used Erlotinib as an EGFR signaling pathway inhibitor. Western blot showed that Erlotinib significantly inhibited EGFR downstream signaling pathway-related protein expression and phosphorylation levels, and subsequently reduced the CCL2 expression in GBM cells (Figure 5D,E). The ELISA assay manifested that Erlotinib markedly reduced the CCL2 secretion under TMZ treatment (Figure 5F). Furthermore, an in vitro migration assay showed that Erlotinib significantly decreased the numbers of migrated macrophages (Figure 5G). Therefore, these findings indicated that ADAM8 induced CCL2 expression to recruit TAMs through the HB-EGF/EGFR signaling pathway.

To validate our observation in vivo, U87MG_scramble and U87MG_shA8 cells were inoculated subcutaneously into the right armpit of BALB/c nude mice (6-week-old, male). Seven days later, the mice bearing tumors received a daily application of 5 mg/kg TMZ for a 5 days on and 2 days off regimen (two cycles in total, intra-tumoral injection). After 28 days, the resulting tumors were removed and measured. The changes in tumor volumes showed that ADAM8 knockdown significantly reduced the tumor growth under TMZ treatment (Figure 5H and Appendix A). U87MG_scramble and U87MG_shA8 cells were orthotopically inoculated into the right frontal lobe of mice for survival analysis. Seven days later, the mice bearing tumors received daily 5 mg/kg TMZ for a 5 days on and 2 days off regimen (two cycles in total, intraperitoneal injection). Consistent with the observed effects on tumor growth in vivo, ADAM8 knockdown markedly prolonged the overall survival of tumor-bearing mice (Figure 5I).

## 4. Discussion

An acquired chemoresistance limits TMZ efficacy in GBM patients. Previous studies have identified ADAM8 as a modulator of chemoresistance in GBM cells. It is commonly recognized that TAMs inhabit GBM tumors with an immunosuppressive pro-tumor phenotype and play pivotal roles for GBM progression [27,28,29,30]. A qPCR screen to detect a spectrum of *ADAM* and *MMP* genes in GBM cells revealed that ADAM8 stands out as gene whose expression is induced by TMZ treatment and further is enhanced by the co-culture with macrophages under TMZ treatment. Consequently, ADAM8 could be a major player for the communication of tumor cells with the tumor microenvironment, in particular in conjunction with TAMs. ADAM8 as an inflammatory mediator regulates diverse pathological processes in CNS inflammation and tumor biology [15,17] through its proteolysis and non-proteolysis function, attributable to the different structural domains present in the full-length protein. In the current study, we demonstrated the roles of ADAM8 in recruiting TAMs to mediate chemoresistance in GBM and simultaneously put forward a potential ADAM8 positive feedback loop involved in the interaction between GBM cells and TAMs under chemotherapy.

ADAM8 overexpression induced by anti-inflammatory macrophages mediates the invasion of pancreatic adenocarcinoma tumor cells [31]. In GBM cells, ADAM8 modulates angiogenesis thereby affecting GBM tumor progression [21,32]. Our previous study manifests that TMZ induced ADAM8 overexpression can mediate TMZ chemoresistance in GBM cells [24]. Hence, we hypothesized that TMZ induced ADAM8 overexpression in GBM cells subsequently modulates the recruitment of TAMs which in turn further enhances TMZ chemoresistance by inducing ADAM8 upregulation in GBM cells in a “malignant positive feedback loop”.

An increasing number of studies investigated how GBM recruit TAMs and maintain an immunosuppressive TME [23,33,34,35]. Various chemokines released from GBM cells attract TAMs directly, whereas some signaling molecules overexpressed in GBM cells induced TAMs recruitment indirectly through regulating the expression of chemokines. In our study, we demonstrated a positive relationship between ADAM8 mRNA expression and Iba-1 staining in human GBM tissues. Moreover, GEPIA data analysis also demonstrated a significant correlation, indicating that ADAM8 may participate in TAMs recruitment. Chemotherapeutic agents can induce TAMs infiltration and orientate TAMs toward tumor-supporting or anti-tumor directions, depending on the type and application schemes of chemotherapeutic agents and the type of tumors [36]. Our results corroborate that TMZ augments macrophage migration in vitro and M2-like macrophages recruitment in vivo in an ADAM8-dependent manner. This is supported by data showing that the knockdown of ADAM8 reduced macrophage migration in vitro and M2-like macrophages recruitment in vivo, indicating a mechanistic role for ADAM8 in TMZ-induced TAMs recruitment in GBM.

The next experimental step was to figure out the mechanism of ADAM8 induced TAMs recruitment. Proteolytic cleavage and non-proteolytic intracellular signal transduction are the two major functions of ADAM8. ADAM8 interacts with integrins through the disintegrin (DIS) domain, thereby activating integrin signaling pathways such as focal adhesion kinase (FAK), extracellular regulated kinase (ERK1/2), and protein kinase B (AKT/PKB) signaling, further contributing to cancer progression via the induction of angiogenesis, metastasis and chemoresistance [24,37]. Through RNA-seq data analysis, we found that HB-EGF was significantly down-regulated in ADAM8 knockdown of U87MG cells. qPCR and western blot were carried out to validate the transcriptional data in U87MG and G1 cells. Moreover, immunofluorescence of human GBM tissues also showed a positive correlation, as there was a stronger staining density of HB-EGF in the high-ADAM8^+^ patients compared to low-ADAM8^+^ patients. the activation of EGFR mediated a variety of intracellular downstream signals, contributing to tumor aggression and resistance to first-line chemotherapies [38,39,40]. Although studies have mentioned that ADAM family proteases can mediate ectodomain shedding of HB-EGF to activate EGFR signaling pathways [22,41], it is still unknown whether ADAM8 is implicated in EGFR activation in GBM under TMZ treatment, in this case by increasing the total amount of HB-EGF, so that shedding of HB-EGF is not the major function of ADAM8 (data not shown). Here, we validated that ADAM8 activates HB-EGF/EGFR signaling pathways in GBM cells as a consequence of TMZ treatment, and ADAM8 knockdown markedly reduced the phosphorylation of EGFR and subsequently the activation of EGFR downstream signals (AKT and ERK signaling).

The regulation of CCL2 expression by ADAM8 through HB-EGF/EGFR signaling is another major finding in our study. CCL2 is a member of the CC chemokine family which acts as a classical chemokine to regulate the chemoattraction of macrophages, monocytes, and other inflammatory cells [23,42,43]. Tumor cells of various types were reported to release CCL2 resulting in the recruitment of macrophages, thereby supporting tumor progression. For instance, Qian et.al demonstrated that CCL2 recruited inflammatory monocytes to facilitate breast-tumor metastasis [44]. Wei et.al reported that the production of CCL2 promoted macrophage recruitment and subsequently colorectal cancer metastasis [45]. Similarly, in several experimental glioblastoma models, tumor cells released CCL2 to attract macrophages [46], and the blockade of CCL2/CCR2 prolonged mouse survival in GBM models [47,48]. Similarly, CCL2-expressing glioma cells induced a 10-fold induction of Ox42-positive cell density in rat models, while tumors overexpressing CCL2 increased more than three-fold, leading to reduced survival in rats [46]. Moreover, Felsenstein et al. showed that TAMs expressed CCR2 to various extents in human GBM specimens and syngeneic glioma models. Glioma inoculation using a Ccr2-deficient strain revealed a 30% reduction of TAMs intratumorally [42]. As An et.al demonstrated that EGFR cooperated with EGFRvIII to induce CCL2-mediated TAMs recruitment in GBM [23], we investigated CCL2 expression in glioblastoma cells under TMZ treatment. Both, in vitro and in vivo, we roundly validated that CCL2 was upregulated in TMZ-treated GBM cells, and ADAM8 knockdown reduced CCL2 expression in GBM cells under TMZ treatment. Furthermore, to demonstrate that CCl2 expression is dependent on EGFR signaling, Erlotinib was used as EGFR signaling pathway inhibitor which significantly reduced CCL2 expression in GBM cells, as shown by western blot and ELISA assays. Our in vitro migration assays showed that Erlotinib significantly decreased the numbers of migrated macrophages. These findings identify an axis in which TMZ induces ADAM8 and leads to downstream signaling that causes the enhanced secretion of CCL2 to recruit TAMs into GBM under TMZ treatment via the HB-EGF/EGFR signaling pathway.

In general, we provide a convincing example that TAMs play pivotal roles in chemoresistance of GBM and further support the point that the tumor microenvironment should be considered as an essential modulator of therapy in GBM. Nevertheless, there are limitations of our subcutaneous immunocompromised models: (i) they don’t reflect the original tumor location, and (ii) they are in immunocompromised mice, which may heighten dependence on innate immunity.

## 5. Conclusions

Taken together, we revealed a novel ADAM8 mediated malignant positive feedback loop between TAMs and GBM cells under TMZ treatment. As such, ADAM8 upregulates the HB-EGF/EGFR signaling-mediated CCL2 expression of GBM cells under TMZ treatment, subsequently inducing TAMs recruitment, which further stimulates ADAM8 upregulation of GBM cells to induce TMZ chemoresistance. These findings support the notion that the tumor microenvironment, in addition to GBM cells and GBM stem-like cells should be considered as an essential modulator of therapy in GBM. Our study provided a theoretical basis for TAM sparing ADAM8-targeting in GBM to optimize standard chemotherapy.

## Figures and Tables

**Figure 1 cancers-14-04910-f001:**
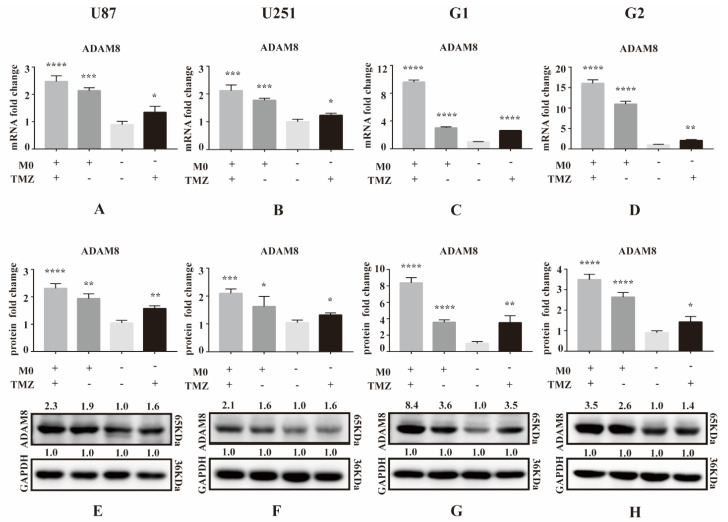
ADAM8 expression induced by TMZ and macrophage. (**A**–**D**) After co-culture with THP-1 derived macrophages for 3 days, mRNA levels of ADAM8 in GBM cell lines U87MG and U251MG and primary cells G1 and G2 with co-culture and TMZ treatment were detected by qPCR. (**E**–**H**) Protein levels of ADAM8 in GBM cell lines U87MG and U251MG and primary cells G1 and G2 with co-culture and TMZ treatment were detected by western blot and measured by Image J. mRNA and protein fold change were statistically shown in the bar graph and data are presented as mean ± SD. TMZ (temozolomide), ADAM8 (A disintegrin and metalloproteinase 8). * *p* < 0.05, ** *p* < 0.01, *** *p* < 0.001, **** *p* < 0.0001.

**Figure 2 cancers-14-04910-f002:**
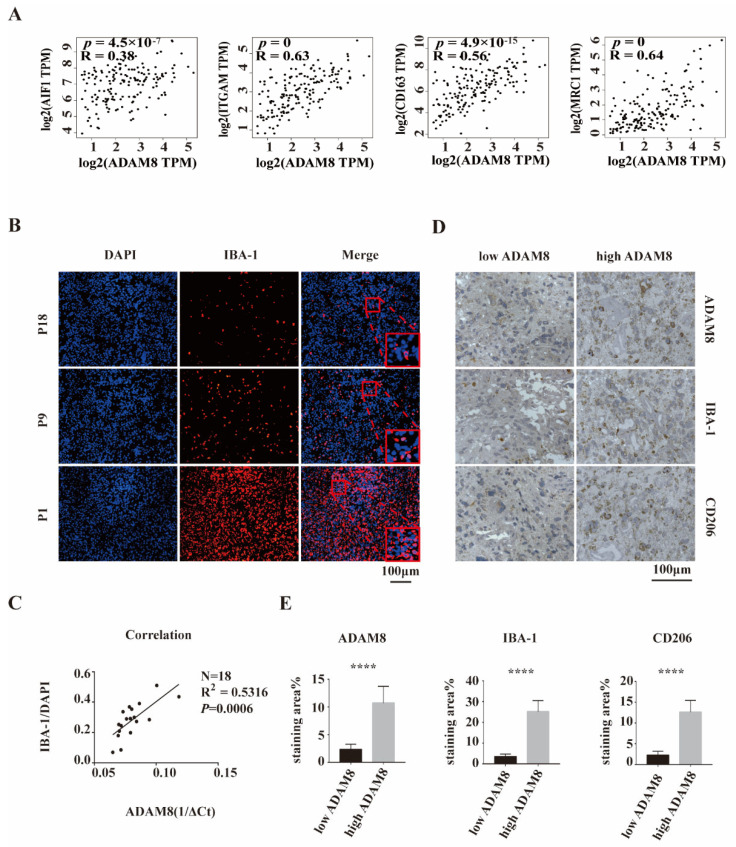
A positive correlation between ADAM8 expression and macrophage infiltration. (**A**) The public dataset GEPIA showed a positive correlation between ADAM8 gene expression and the expression of TAMs signatures including Iba-1, CD11b, CD163, and CD206. (**B**) Iba-1 staining (red) of our GBM cohort patients (P1, P9, and P18) was detected by immunofluorescence, using DAPI for nuclei labeling. (**C**) Positive correlation between mRNA levels of ADAM8 and density of Iba-1 positive cells in 18 human GBM tissues was verified by qPCR (*N* = 18, R^2^ = 0.5316, *p* = 0.0006). (**D**) Immunohistochemistry staining of ADAM8, Iba-1, and CD206 of GBM tissues in two defined groups (low ADAM8 and high ADAM8) using the median value of ADAM8 expression as classification criteria. (**E**) The bar graph showed the statistical results of immunohistochemistry staining of ADAM8, Iba-1, and CD206, measured by Image J and analyzed by an unpaired two-tailed Student’s *t*-test. scale bar = 100 um. **** *p* < 0.0001.

**Figure 3 cancers-14-04910-f003:**
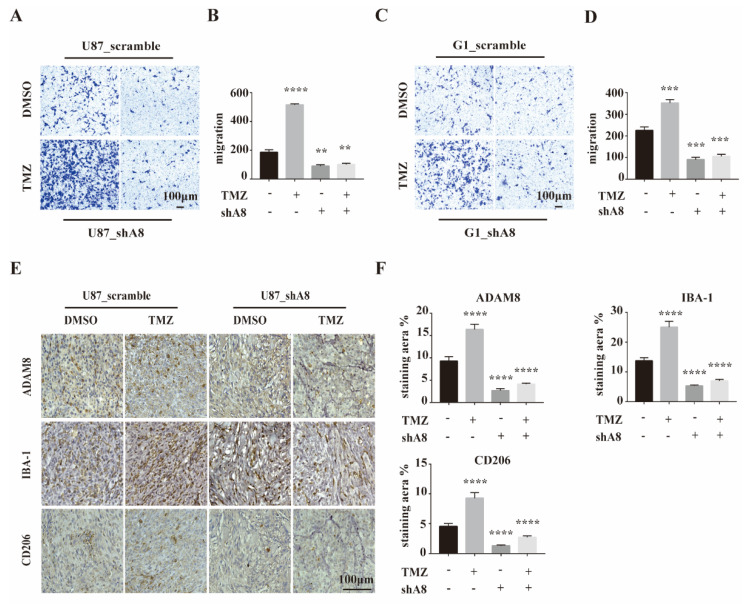
ADAM8 induced macrophage recruitment in vitro and in vivo. (**A**,**C**) THP-1 derived macrophages co-culturing with the indicated GBM cell lines with ADAM8 knockdown and TMZ treatment. Migration activity was detected by transwell assay in vitro. (**B**,**D**) The bar graph represented the statistical results of migrated macrophages. (**E**) Representative immunohistochemistry staining of ADAM8, Iba-1, and CD206 of subcutaneous inoculation models. (**F**) The bar graph represented the statistical results of the immunohistochemistry staining of ADAM8, Iba-1, and CD206, respectively. All experiments were repeated at least three times, and data were analyzed using a student’s *t*-test. scale bar = 100 um. shA8 represents ADAM8 knock down. ** *p* < 0.01, *** *p* < 0.001, **** *p* < 0.0001.

**Figure 4 cancers-14-04910-f004:**
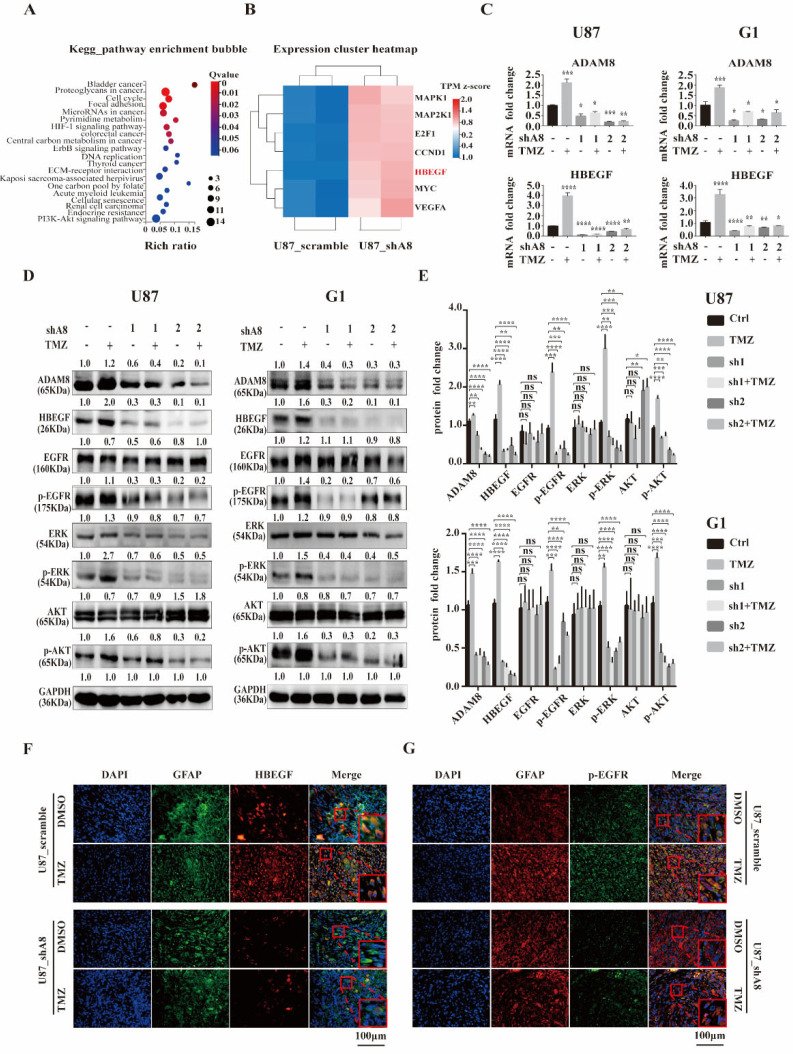
ADAM8 regulates HB-EGF/EGFR signal pathway. (**A**) Enrichment bubble figure indicates down-regulated genes of RNA sequencing of U87MG_scramble samples and U87MG_shA8 samples, which are involved in specific signaling cascades. (**B**) Heatmap denotes the signatures enriched in “Bladder cancer” pathways. (**C**) mRNA levels of ADAM8 and HB-EGF in indicated GBM cell lines (U87MG and G1). (**D**) Protein levels of ADAM8, HB-EGF, and EGFR signaling cascades in indicated GBM cell lines (U87MG and G1). (**E**) Protein levels were measured by Image J and are statistically shown in the bar graph using an unpaired two-tailed Student’s *t*-test. (**F**,**G**) Immunofluorescence staining of HB-EGF and p-EGFR of xenograft tissue sections, respectively. Scale bar = 100 um. ns (non-significant) * *p* < 0.05, ** *p* < 0.01, *** *p* < 0.001, **** *p* < 0.0001.

**Figure 5 cancers-14-04910-f005:**
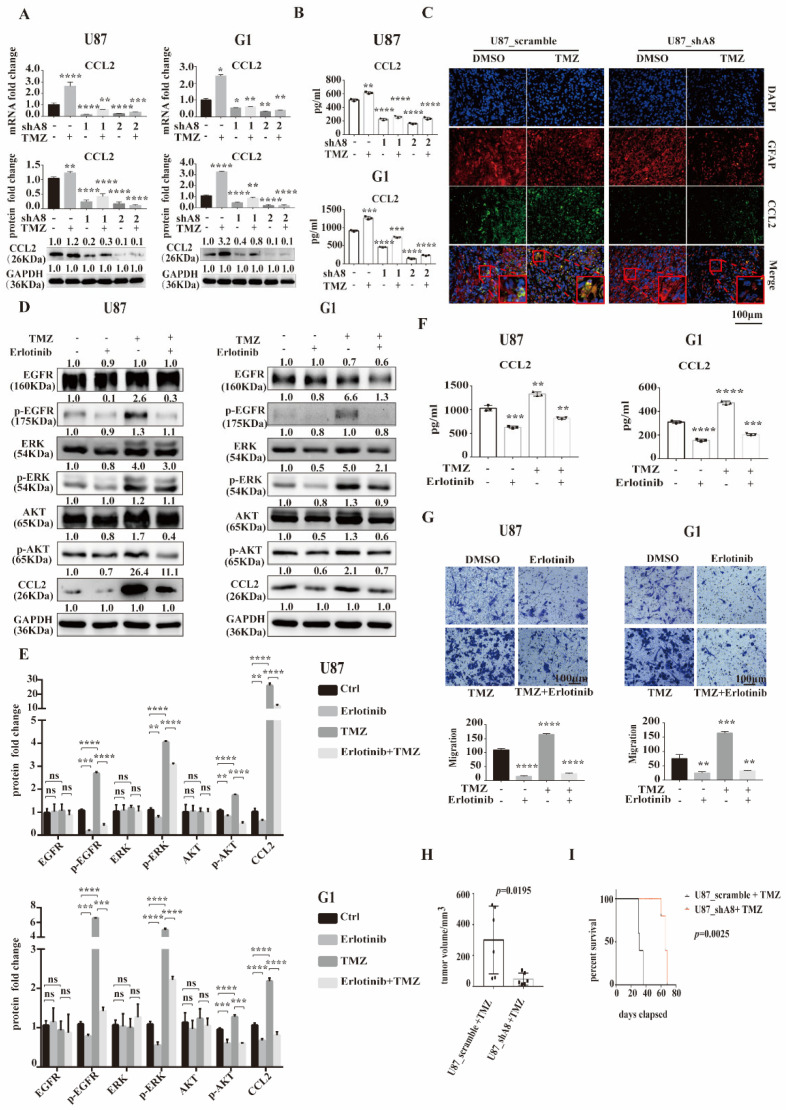
ADAM8 induces HB-EGF/EGFR mediated CCL2 expression. (**A**) mRNA and protein levels of CCL2 in indicated GBM cell lines (U87MG and G1). (**B**) Secreted CCL2 levels in the supernatant of indicated GBM cell lines with ADAM8 knockdown and TMZ treatment were detected by ELISA (U87MG and G1). (**C**) Immunofluorescence staining of CCL2 in xenograft tissue sections. (**D**) Western blot showed the protein levels of EGFR signaling cascades and CCL2 in the indicated GBM cell lines (U87MG and G1) in the presence of TMZ and Erlotinib. (**E**) Protein fold changes were statistically shown in the bar graph using an unpaired two-tailed Student’s *t*-test (**F**) Secreted CCL2 levels in the supernatant of indicated GBM cell lines in the presence of TMZ and Erlotinib were detected by ELISA (U87MG and G1). (**G**) Motility of THP-1 derived macrophages co-culturing with the indicated GBM cell lines with TMZ and Erlotinib treatment were detected by transwell assay in vitro and were statistically shown in the bar graph. (**H**) Tumor volumes of subcutaneous U87MG xenograft models were statistically shown in the bar graph (*N* = 6, *p* = 0.0195). (**I**) Survival curve of in-site inoculation U87MG models (*N* = 5, *p* = 0.0025) scale bar = 100 um. * *p* < 0.05, ** *p* < 0.01, *** *p* < 0.001, **** *p* < 0.0001.

## Data Availability

The datasets used and/or analyzed during the current study are available from the corresponding author on reasonable request.

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
