# Peer review of "The GBM Tumor Microenvironment as a Modulator of Therapy Response: ADAM8 Causes Tumor Infiltration of Tams through HB-EGF/EGFR-Mediated CCL2 Expression and Overcomes TMZ Chemosensitization in Glioblastoma"

_cancers, 2022, doi:10.3390/cancers14194910_

Round 1

Reviewer 1 Report

Major points:

1.                  The authors monitor the gene expression of the studied genes at both mRNA and protein levels. While expression at the mRNA level is quantified and statistical differences are made, protein expression is given for guidance only. This is a fundamental mistake! Proteins but not mRNA govern the phenotypic manifestation /function! Changes in protein expression must be accompanied by a quantitative evaluation (Fig. 1e-h, 4d, 5a, d)!

2.                  Can you demonstrate more directly the resistance to TMZ in GMB cells?

Minor points:

1.                  The descriptions of the Figures are not easy to understand. All relevant information should be listed here.

2.                  All abbreviations used in Figure descriptions should be explained.

Author Response

Reviewer 1

Comments and Suggestions for Authors

Major points:

  1. The authors monitor the gene expression of the studied genes at both mRNA and protein levels. While expression at the mRNA level is quantified and statistical differences are made, protein expression is given for guidance only. This is a fundamental mistake! Proteins but not mRNA govern the phenotypic manifestation /function! Changes in protein expression must be accompanied by a quantitative evaluation (Fig. 1e-h, 4d, 5a, d)!

Response: We thank for the suggestion. Indeed, as suggested by the reviewer, we have now added quantification of our WB blots (see Fig.1e-h, 4d, 5a, d) for verification of our mRNA results.

  1. Can you demonstrate more directly the resistance to TMZ in GMB cells?

Response: As suggested by the reviewer, we have now added multiple experiments regarding GBM co-culture with macrophages in the presence of TMZ (see Supplementary Figure 2). We observed that macrophage promotes GBM in vitro proliferation, migration and invasion in the presence of TMZ, indicating that macrophage induces GBM chemoresistance in vitro. We have now described the co-culture schemes in the method section “2.6. Cell proliferation assay (line 159) and “2. 7. Cell migration and invasion assay” (line 168) and the effect in the result part “3.3. ADAM8 induces macrophage recruitment in vitro and in vivo(line 308)

Minor points:

  1. The descriptions of the Figures are not easy to understand. All relevant information should be listed here.

Response: Thanks for your kind advices. We have now checked all the figure legends and made it easy for reader to understand

  1. All abbreviations used in Figure descriptions should be explained.

Response: Thanks for your suggestion. We have now explained all abbreviations in the figures.

Reviewer 2 Report

The authors describe an interesting study showing that the tumour microenvironment in GBM, a poor outcome cancer,  plays a significant role in disease progression in immune suppression. In particular, they show that the metalloproteinase ADAM8, which is upregulated in GBM and plays a role in stromal remodelling, also regulates macrophage infiltrate in GBM through upregulation of CCL2 in GBM cells. Infiltrating macrophages that are recruited can further stimulate ADAM8 expression in GBM cells. This work builds on the authors previous findings that ADAM8 upregulation in GBM was linked to temozolomide (GBM first-line chemotherapy) resistance. Thus the findings imply that tumour associated macrophage recruitment can further enhance the ADAM-8 linked mechanism of temozolomide resistance.

I have only the following minor issues that need to be addressed:

1.       Page4, method 2.9: authors mention that temozolomide was administrated “about 7 days later…”. Does this mean timing was not standard post implant for all mice? What was the basis of timing for TMZ administration? This needs ot be clarified.

2.       Figure 1: It would be easier to read the graphs if the double negative control was on the left hand column (first column) and the double treatment was on the right (final column) of each graph.

3.       It would be good to see a brief discussion on the limitation of the sub-cutaneous models used as (i) they don’t reflect the original tumour location and (ii) they are in immunocompromised mice, which may heighten dependence on innate immunity. The inclusion of an orthotopic model alleviates location concerns, but this should be briefly addressed.

4.       Figure 2: many of the y-axes have unclear labels e.g. for panel E, what does “Area%” mean?

5.       Figure 4E-F: Need magnified views of immunofluorescence.

6.       Minor spelling and grammatical errors throughout e.g. pg 10, “3.5. ADAM 8 induce HB-EGF…”, use of the term “manifested” should be replaced with straightforward terminology e.g. “showed that”

7.       The orthotopic results in Figure S3 should be in the main figures as this is a critical finding for the study. If feasible, TAMs should be measured in this model.

8.       Figure 5F: Does “count” on the y-axis refer to the number of macrophages on the internal side of the transwell membrane? Was it standardised to total cell count (sum of both sides)?

Author Response

Reviewer2

Comments and Suggestions for Authors

The authors describe an interesting study showing that the tumour microenvironment in GBM, a poor outcome cancer, plays a significant role in disease progression in immune suppression. In particular, they show that the metalloproteinase ADAM8, which is upregulated in GBM and plays a role in stromal remodelling, also regulates macrophage infiltrate in GBM through upregulation of CCL2 in GBM cells. Infiltrating macrophages that are recruited can further stimulate ADAM8 expression in GBM cells. This work builds on the authors previous findings that ADAM8 upregulation in GBM was linked to temozolomide (GBM first-line chemotherapy) resistance. Thus the findings imply that tumour associated macrophage recruitment can further enhance the ADAM-8 linked mechanism of temozolomide resistance.

I have only the following minor issues that need to be addressed:

  1. Page4, method 2.9: authors mention that temozolomide was administrated “about 7 days later…”. Does this mean timing was not standard post implant for all mice? What was the basis of timing for TMZ administration? This needs ot be clarified.

Response: Many thanks for your questions. In this study, temozolomide are administrated at the same timing (7 days after implanting tumor cells) to all mice. After reading related references, the basis of timing for TMZ administration depends on when implanting cells grow into evident tumor bulk. in this case, we can see evident subcutaneous tumor bulk for all mice after 7 days. In terms of orthotopic models, Kai Huang et al. described in their article “Genome‐Wide CRISPR‐Cas9 Screening Identifies NF‐κB/E2F6 Responsible for EGFRvIII‐Associated Temozolomide Resistance in Glioblastoma” that TMZ was administrated one week (7 days) later [1]. However, Xiao-Ning Zhang et al. described that TMZ was administrated two weeks (14 days) later [2]. To avoid this confusion, I have now removed the word “about” in the revised manuscript.

  1. Figure 1: It would be easier to read the graphs if the double negative control was on the left hand column (first column) and the double treatment was on the right (final column) of each graph.

Response: Thanks for your professional advices. Yes, i totally agree with your point. However, picture of Western blot band can’t be rearranged the way you advised due to my ignorance of this common rules at the beginning. I want to keep the Figure1 unchanged if you don’t insist repeating western blot experiment, considering only 10 days left for me to submit the revised manuscript. I hope with all my heart that it is ok for the reviewer. Of course, I will repeat the western blot experiment if you insist. Thanks for your kind and professional advices again!

  1. It would be good to see a brief discussion on the limitation of the sub-cutaneous models used as (i) they don’t reflect the original tumour location and (ii) they are in immunocompromised mice, which may heighten dependence on innate immunity. The inclusion of an orthotopic model alleviates location concerns, but this should be briefly addressed.

Response: Many thanks for your kind and professional advices. A brief discussion on the limitation of the sub-cutaneous models has been added in the manuscript as follows:

“In general, we provide a convincing example that TAMs play pivotal roles in chemoresistance of GBM and further support the notion that tumor microenvironment, in addition to GBM cells and GBM stem-like cells, should be considered as an essential modulator of therapy in GBM. Nevertheless, there are limitations of our subcutaneous immunocompromised models: (i) they don’t reflect the original tumour location, and (ii) they are in immunocompromised mice, which may heighten dependence on innate immunity.”

  1. Figure 2: many of the y-axes have unclear labels e.g. for panel E, what does “Area%” mean?

Response: Thanks for your kind advices. I have checked all the unclear y-axes labels in all the figures and replaced them with a clear label or explained them in the figure legends. In panel E of Figure 2, “Area%” means the positive staining area of ADAM8, IBA-1 and CD206 in panel D, analyzed by Image J.  

  1. Figure 4E-F: Need magnified views of immunofluorescence.

Response: Many thanks for your kind advices. Magnified pictures of Figure4E-F have been shown as Figure4. Besides, other immunofluorescence pictures have been magnified for a clearer viewing.   

  1. Minor spelling and grammatical errors throughout e.g.pg 10, “3.5. ADAM 8 induce HB-EGF…”, use of the term “manifested” should be replaced with straightforward terminology e.g “showed that”

Response: Many thanks for so careful viewing. The spelling and grammatical errors were corrected as you suggested.

  1. The orthotopic results in Figure S3 should be in the main figures as this is a critical finding for the study. If feasible, TAMs should be measured in this model.

Response: Thanks for your professional advices. The graph of tumor volume and survival curve have been included in the main figures as you suggested. Unfortunately, we did not repeat all our findings in the orthotopic models, which has been discussed as limitations as you kindly suggested.

  1. Figure 5F: Does “count” on the y-axis refer to the number of macrophages on the internal side of the transwell membrane? Was it standardised to total cell count (sum of both sides)?

Response: “count” on the y-axis refers to the number of macrophages on the bottom side of the chamber, which was stained and showed as pictures. It was not standardized to total cell count since we seed the same number of cells in each well. Although there are references described the same methods [3,4], I must agree that it will be more convincing if it is standardized to total cell count as you kindly raised.  

  1. Huang, K.; Liu, X.; Li, Y.; Wang, Q.; Zhou, J.; Wang, Y.; Dong, F.; Yang, C.; Sun, Z.; Fang, C.; et al. Genome-Wide CRISPR-Cas9 Screening Identifies NF-kappaB/E2F6 Responsible for EGFRvIII-Associated Temozolomide Resistance in Glioblastoma. Adv Sci (Weinh) 2019, 6, 1900782, doi:10.1002/advs.201900782.
  2. Zhang, X.N.; Yang, K.D.; Chen, C.; He, Z.C.; Wang, Q.H.; Feng, H.; Lv, S.Q.; Wang, Y.; Mao, M.; Liu, Q.; et al. Pericytes augment glioblastoma cell resistance to temozolomide through CCL5-CCR5 paracrine signaling. Cell Res 2021, 31, 1072-1087, doi:10.1038/s41422-021-00528-3.
  3. Pan, S.; Shen, M.; Zhou, M.; Shi, X.; He, R.; Yin, T.; Wang, M.; Guo, X.; Qin, R. Long noncoding RNA LINC01111 suppresses pancreatic cancer aggressiveness by regulating DUSP1 expression via microRNA-3924. Cell Death Dis 2019, 10, 883, doi:10.1038/s41419-019-2123-y.
  4. Zhou, Y.; Wang, Y.; Wu, S.; Yan, Y.; Hu, Y.; Zheng, Z.; Li, J.; Wu, W. Sulforaphane-cysteine inhibited migration and invasion via enhancing mitophagosome fusion to lysosome in human glioblastoma cells. Cell Death Dis 2020, 11, 819, doi:10.1038/s41419-020-03024-5.

Reviewer 3 Report

An interesting manuscript by Liu et al on the GBM tumour microenvironment as a modulator of therapy response.

Comments:

- Line 178, human U87MG should be Human U87MG.

- Lines 243-250, more details are needed for the data used in the GEPIA software for the analysis of results and suggestive conclusions.

- Line 282, Figure 2B images appear rather dark so difficult to view results easily. Perhaps the authors would like to consider using better quality images for their paper.

-Regarding the tumour formation in Supplementary Figure S3, the pictures of the mice tumours between U87-MG scramble and U87-MG shA8 appear quite similar overall and though the authors have shown a significant reduction in tumour volume is not depicted by the photos showing extracted tumours but they are evident from the mice. The authors need to re-check these pictures.

Author Response

Reviewer 3

Comments and Suggestions for Authors

An interesting manuscript by Liu et al on the GBM tumour microenvironment as a modulator of therapy response.

Comments:

1 Line 178, human U87MG should be Human U87MG.

Response Many thanks for so careful observation, I have corrected the spelling

2 Lines 243-250, more details are needed for the data used in the GEPIA software for the analysis of results and suggestive conclusions.

Response: Thanks for your professional advices. I have added some details (GBM samples and TCGA projects) as written in the manuscript:

“Bioinformatic analyses of correlation using the public dataset GEPIA (Gene Expression Profiling In-teractive Analysis) were performed. We select GBM samples from the TCGA projects on the GEPIA and we found a positive correlation between ADAM8 gene expression and expression of TAM signatures including Iba-1 (AIF1, R=0.38, p=4.5 x 10-7), CD11b (ITGAM, R=0.63, p=0), CD163 (R=0.56, p=4.9 x 10-15), and CD206 (MRC1, R=0.64, p=0) (Figure 2A), indicating that ADAM8 expression is associated with TAMs and may play a role for attracting TAMs into GBM.”

3 Line 282, Figure 2B images appear rather dark so difficult to view results easily. Perhaps the authors would like to consider using better quality images for their paper.

Response: Thanks for your suggestions. the figure2B images have been replaced with better quality images and magnified viewing, I hope the new images are fine for you to view.

4 Regarding the tumour formation in Supplementary Figure S3, the pictures of the mice tumours between U87-MG scramble and U87-MG shA8 appear quite similar overall and though the authors have shown a significant reduction in tumour volume is not depicted by the photos showing extracted tumours but they are evident from the mice. The authors need to re-check these pictures.

Response: Many thanks for your kind advices. I have re-checked these pictures and I am sure that the extracted tumours are from the mice. Honestly, I made a mistake of taking photos. I shouldn’t have taken photos of the two groups separately. Considering the issue, so I put this pictures in the supplementary figures. As you can see from the blue diamond background of the extracting tumor photos, the photo of U87-MGshA8 was more magnified than the U87-MG scramble group (because I want to layout these pictures more even). Thanks to your careful viewing, I have corrected this issue. Many thanks again !   

Round 2

Reviewer 1 Report

The authors have dealt with all my comments. I think the revised MS is suitable for publication in the Cancers.

Author Response

Thank you so much for your kind and professional comments!